# The Reasonableness and Spatial Differences of the Food Consumption Structure of Urban and Rural Residents in China, 2015–2021

**DOI:** 10.3390/foods12101997

**Published:** 2023-05-15

**Authors:** Yanyan Lian, Lijuan Gu, Linsheng Yang, Li Wang, Hairong Li

**Affiliations:** 1Key Laboratory of Land Surface Pattern and Simulation, Institute of Geographical Sciences and Natural Resources Research, Chinese Academy of Sciences, Beijing 100101, China; 2College of Resources and Environment, University of Chinese Academy of Sciences, Beijing 100049, China

**Keywords:** food consumption, balanced dietary pattern, urban-rural differences, spatial differences, China

## Abstract

Based on residents’ food consumption data from 31 provinces in China from 2015–2021, this study analyzes the deviation in food consumption from nutrition targets and the spatial distribution characteristics of urban and rural residents in China from 2015–2021, and finds that there are irrationalities in the structure of food consumption of Chinese residents as well as regional differences in consumption. The food consumption of Chinese residents deviates from the recommended values of the Chinese Food Guide Pagoda to a certain extent, with large differences between urban and rural areas and provinces. Therefore, a new concept of food security with nutrition as the target should be established to guide residents’ food consumption scientifically and rationally, and to adopt focused attention and targeted measures for regions with serious imbalances in food consumption.

## 1. Introduction

At present, China’s rapid socio-economic development has led to significant changes in the structure of residents’ food consumption. In the past, direct consumption such as rations dominated the food consumption of Chinese residents, while indirect food consumption such as meat, poultry, eggs, and milk was relatively small. Studies [1,2] have shown that the consumption of food rations has been declining more significantly in the last three decades, while the consumption of other food categories has shown a rapid growth. The structure of animal-based and plant-based foods has changed significantly since the 1980s to 2015 [3]. The current food consumption demands of residents have shifted from initial subsistence to diversification [4].

However, the dietary structure has irrational problems, with low levels of consumption of fruits and vegetables, milk, and fish and shrimp [5]. Moreover, China is a vast country with complex topography and diverse climate, and there are obvious differences in dietary habits and food consumption in different regions [6,7]. In addition, it is often statistically divided into two parts: urban and rural, with urban referring to the residential councils and other areas connected to the actual construction of municipal, district, and county government sites in municipal districts and unincorporated cities, and rural being the areas outside of towns [8]. There is also an unbalanced development of dietary structure in urban and rural areas [9].

Food consumption is an important indicator of national living standards, and it is also an important basis for policy formulation in agricultural production restructuring, market circulation, and nutrition improvement. Food systems, human health, and the environment are interconnected. The food we eat and the way we produce it will determine the health of humans and the planet, and major changes must be made in response to existing problems in order to avoid a reduction in life expectancy and continued environmental degradation [10,11,12,13]. In terms of human health, the Chinese Dietary Guidelines is a visual representation of dietary health. It follows the principle of a balanced diet and divides the pagoda into five layers with different sizes to reflect the five major food groups and the amount of food. These five major food groups are cereals and potatoes; vegetables and fruits; livestock, poultry, fish, and eggs; milk, soybeans, and nuts; and cooking oil and salt.

Many scholars have conducted different types of studies on food consumption; descriptions and aspects of these follow. The first type of study is on changes in food consumption, which often use long-term time-series data on food consumption to elaborate on the trends and patterns of changes in the food consumption structure of China at the macro level [1,5]. The second type of study focuses on forecasting future food demand and supply for the whole country or for a key region using methods such as formula calculations [14,15]. The third type of study is based on a nutritional perspective and investigates the characteristics of food consumption and dietary nutrition status and the differences between them by methods such as household surveys or macro statistics [2]. The fourth category in the literature examines interactions between food consumption and various types of resources and environments. The last category studies socioeconomic factors, such as income, urbanization level, food price index, and food production, that have an impact on residents’ food consumption [16]. These factors can, in turn, influence the dietary structure of the residents by changing their health consciousness, dietary preferences, ability to access food, and ease of access to food.

Based on the existing literature, this study explores the main characteristics and problems of current food consumption and nutritional development of Chinese urban and rural residents and the influencing factors. The latter work is based on the theory of dietary balance, and analyzes the differences between geographical areas to see if problems in food consumption have been solved or improved in the last seven years compared to the past.

## 2. Materials and Methods

### 2.1. Data

#### 2.1.1. Food Consumption Data

Food consumption data from urban and rural residents in 31 provinces in China (excluding Hong Kong, Macao, and Taiwan) during 2015–2021 are from the China Yearbook of Household Survey (2016–2022).

The China Yearbook of Household Survey is a national survey of household income, expenditure, and living conditions in order to provide a comprehensive, accurate, and timely understanding of national and regional income, consumption and other living conditions, and quality of life of urban and rural residents. The household survey is conducted on a provincial basis using a stratified, multi-stage, random probability sampling method proportional to the size of the population and a uniform questionnaire to collect the survey content, which is finally aggregated and calculated by the National Bureau of Statistics [17].

The food groups covered in this paper include grain (including cereals), edible oil and fats, vegetables, meat, poultry, aquatic products, eggs, milk and dairy products, and fruits. For the sake of analysis, some of the food items were grouped later, with vegetables and fruits grouped under the category of fruits and vegetables, and meat, poultry, aquatic products, and eggs grouped under the category of animal-based foods.

#### 2.1.2. Demographic Data

Urban and rural population data by province in 2021 were from the China Statistical Yearbook.

#### 2.1.3. Influencing Factor Data

Data by province on urbanization rate, food price index (grain, edible oil and fats, vegetables, meat, poultry, aquatic products, eggs, milk and dairy products, fruits), disposable income of urban and rural residents, per capita output of food (grain, edible oil and fats, meat, aquatic products, milk) from 2015–2021 were from the China Statistical Yearbook.

### 2.2. Methods

#### 2.2.1. The Food Consumption Structure

The calculation of the per capita consumption of various types of food as a percentage of total food consumption for urban and rural residents in China from 2015 to 2021 is as follows:P_i_ = C_i_/∑C_i_ × 100%(1)
where C represents the actual consumption of food; i represents each type of food.

The ratio of rural to urban consumption of food types for 2015–2021 is calculated with the following formula:P_r/u_ = Cr_i_/Cu_i_(2)
where C_r_ represents the actual consumption of food by rural residents, C_u_ represents the actual consumption of food by urban residents, and i represents each type of food.

#### 2.2.2. Balanced Diet Situation

The Chinese Dietary Guidelines was proposed to address the nutritional status and dietary structure characteristics of Chinese residents, in order to help them make rational food choices and to reduce or prevent the occurrence of chronic diseases.

The food consumption data in 2021 were compared to the minimum and maximum standards of food intake recommended in Chinese Food Guide Pagoda (2022) to analyze the deviation in food consumption of the population from the recommended standards. The recommended minimum and maximum values are shown in Table 1 [18]. Not all foods in the dietary pagoda were compared because data were not available for some food groups.

The formula for the deviation is as follows:D_i_ = C_i_/RC_i_ × 100%(3)
where C represents the actual consumption of food, RC represents the recommended value of the dietary pagoda, and i represents the food groups.

#### 2.2.3. Spatial Autocorrelation Analysis of Food Consumption

GeoDa 1.20 software was applied to determine whether food consumption in neighboring areas would show convergence. A distance-based spatial weighting matrix was used and Moran’s I index was calculated. The Moran’s *I* index takes values in the range [−1, 1], with *I* > 0 being positive correlation, and larger values indicate greater correlation in the spatial distribution, i.e., greater spatial aggregation. For the statistical test of Moran’s *I*, the test statistic Z of the approximate normal distribution under random conditions was used, and differences were considered statistically significant if *p* < 0.05. A local autocorrelation analysis was then performed to describe the correlation between each province and its neighboring provinces in terms of food consumption and to identify specific areas of aggregation. The results were divided into four types of regions: high-high aggregation, high-low aggregation, low-high aggregation, and low-low aggregation. Finally, ArcGIS 10.5 software was used for visual presentation.

#### 2.2.4. Regional Balanced Diet Situation

China is divided into several regions which were used to compare the differences in food consumption and the deviation from the nutrition targets among residents in different regions of China in 2021. The division is based on seven economic collaboration zones [19], which are able to express certain geographical and economic differences, as shown in Figure 1.

The formula for regional per capita food consumption is as follows:A_i_ = ∑(C_i_ × P_j_)/∑P_j_(4)
where C represents actual consumption of food, P represents population size, i represents food groups, and j represents individual provinces within a large region.

#### 2.2.5. Analysis of Influencing Factors

A linear regression analysis of the factors influencing food consumption was conducted using SPSS software. The dependent variables were the per capita consumption of grain (Y_1_), edible oils and fats (Y_2_), vegetables (Y_3_), meat (Y_4_), poultry (Y_5_), aquatic products (Y_6_), eggs (Y_7_), milk and dairy products (Y_8_) and fruits (Y_9_) by urban and rural residents in each province during 2015-2021. The independent variables included urbanization rate (X_1_), urban and rural per capita disposable income (X_2_), and consumer price index (X_3_), and production (X_4_) of the corresponding food groups (Since meat and poultry are not counted separately in the yearbook in 2015, the same data are used here). The selection of variables is shown in Table 2.

## 3. Results

### 3.1. Changes in the Food Consumption Structure

The change in the structure of food consumption is shown in Figure 2. The share of per capita consumption of cereals showed an overall downward trend. In particular, the proportion of cereals consumption by rural residents fell from 44.8% to 38.8%, showing a faster downward trend, while the consumption of urban residents changed little. Compared to urban residents, the consumption of animal-based foods, fruits and vegetables by rural residents increased more significantly.

In recent years, the ratio between rural and urban residents increased for all food groups except grain, for which the difference between urban and rural areas gradually decreased. In 2021, the rural/urban ratios for aquatic products, milk and dairy products, fruits were 0.65, 0.51, and 0.77, respectively, and are still likely to increase. While the ratios for vegetables, meat, poultry and eggs were all close to 1.

### 3.2. Rationality of Food Consumption

The structure of food consumption has become more diverse in recent years. However, from the perspective of nutrition and health, residents should not only eat enough, but also eat healthily. The comparison of food consumption by Chinese provinces in 2021 to the minimum and maximum standards recommended by the Chinese Food Guide Pagoda (2022) is shown in Figure 3 and Figure 4.

Cereals was the only food type for which the actual consumption in 2021 exceeded the minimum recommended by the Dietary Guidelines in all provinces. Meanwhile, all other food groups were under-consumed in individual provinces. However, rural residents consumed too much cereals, exceeding the recommended maximum (300 g) in all provinces. In terms of animal-based foods, there were differences in consumption between regions. Rural residents in the western region (including Guizhou, Tibet, Shaanxi, Gansu, Qinghai, Ningxia, and Xinjiang) consumed less than the minimum recommended standard, while the more economically developed eastern regions, such as Shanghai, Zhejiang, and Guangdong, consumed more than 30% of the maximum recommended standard for animal-based foods. Edible oil and fats consumption exceeded the maximum recommended standards to a greater extent in rural areas than in urban areas, especially in Hubei, Hunan, Chongqing, Jiangxi and Heilongjiang, where local residents like to eat spicy food or have heavy diets. The largest deviation between consumption and recommended intakes was for milk and dairy products, with all regions failing to meet the recommended standard and the highest value reaching only 29.0% of the minimum recommended value. This is followed by fruits and vegetables, with the Qinghai-Tibet Plateau having the most serious deviation; urban and rural fruit consumption in Tibet reached only 35.9% and 8.5% of the minimum recommended standard, respectively.

The trend in food consumption as a percentage of the dietary pagoda from 2015–2021 is shown in Figure 5. It shows that the trend in cereals consumption varies across provinces, with only a few provinces (Tianjin, Inner Mongolia, Guangdong, Tibet and Xinjiang) where not only the percentage of consumption is decreasing, but also the consumption is decreasing. The trend in consumption of vegetables and fruits is increasing or fluctuating in most areas, but has not yet reached the recommended intake. Consumption of animal-based foods is increasing the fastest of all the food groups (with the exception of rural areas in Tibet). It is worth noting that, in contrast to other food groups, the change in milk and dairy products consumption was not significant, with per capita consumption of 46.8 and 17.3 g/day for urban and rural residents, respectively, in 2015, increased to 49.9 and 25.5 g/day, respectively, in 2021, which represents a very low growth rate. The low intake of milk and dairy products has long been a prominent problem in the food consumption. In terms of edible oil consumption, growth was insignificant or started to decline in urban areas in most provinces, but grew in rural areas.

### 3.3. Food Consumption Cluster

There were obvious spatial differences in the per capita consumption of major food in China, and this paper conducted spatial autocorrelation analysis on several major food groups to determine whether there was a convergence in food consumption in neighboring regions.

The Moran’s *I* index for per capita consumption of major food items in 2021 was calculated (Table 3). Except for cereals, edible oil and fats, meat, the Moran’s *I* indices of food consumption in all other categories were positive, indicating that there is positive spatial autocorrelation in the per capita consumption of major food items among Chinese residents.

The LISA clustering map of per capita food consumption in 2021 was drawn, as shown in Figure 6 and Figure 7. The results showed that there is a clear regional clustering of consumption of each food group. High-high aggregation areas for poultry are concentrated in the southeast and low-low aggregation areas in the north; high-high aggregation areas for aquatic products consumption are also in the southeast and low-low aggregation areas in the west of China; high-high aggregation areas for eggs, dairy and fruits are in the south of China and low-low aggregation areas are in the central and north.

### 3.4. Regional Food Consumption

There was an obvious geographical clustering of consumption of different types of food in China. This paper compared the differences in food consumption and the deviation from the nutrition targets in different regions in China in 2021, based on the division into seven geographical regions in China, and the results are shown in Figure 8.

In 2021, the average daily consumption of cereals in each of the seven regions far exceeds the minimum recommended value of the dietary pagoda, and rural inhabitants consume too much. The residents in southern China and northwestern China had the greatest under-consumption of vegetables. In terms of fruits consumption, only urban residents in northern China and northeastern China barely met the standard. Except for rural residents in the northwestern China, all regions were able to meet the minimum standard of animal-based foods intake, with the eastern China and southern China having the most adequate intake of animal food, which exceeded the maximum standard. Milk and dairy products are well below the lower limit of the recommended range in both urban and rural areas or in either region. In terms of edible oil and fats consumption, the recommended standard was largely met in rural and urban, while the exceedance of the standard was serious in rural areas.

### 3.5. Influencing Factors of Food Consumption

Food consumption is influenced by multiple factors. Figure 9 shows that there is a significant relationship between food and urbanization rate. Grain and edible oils and fats were negatively correlated, while vegetables and fruits were positively correlated with urbanization rate; these findings are consistent with the higher consumption of food and edible oils and fats in rural areas and better vegetables and fruits eating habits in urban areas, as mentioned above. Disposable income significantly influences the consumption of meat. Per capita food production also has a strong impact on food consumption, especially on aquatic products and dairy products. Inner Mongolia has the highest milk production [20] and, accordingly, local consumption of dairy products is higher in this region. The same is true for aquatic products consumption in the southeast coast.

## 4. Discussion

Food consumption among Chinese residents is now a diversified structure dominated by grain consumption and supplemented by meat, fruits, vegetables, aquatic products, poultry, eggs, milk, and dairy products. Hou et al. studied the changes in dietary consumption of the Chinese population between 1980 and 2021, which were mainly characterized by a decline in direct consumption of cereals, a rapid rise in consumption of animal products, and a continuous increase in vegetables and fruits [1]. The results presented in this paper are consistent with that study. In recent years, the standard of living in rural areas has continued to improve and the supply of agricultural products in the market has become more abundant, giving rural residents a wider choice of food types, a trend that is still present in the food consumption of rural residents. The change in the structure of food consumption is more pronounced among rural residents than among urban residents. However, this change has slowed down in urban areas and is approaching equilibrium, with the rate of change remaining more or less the same over the past seven years.

As the economy and society continue to develop, the consumption of foods in the upper levels of the Chinese Food Guide Pagoda has increased, and the consumption of foods in the lower levels has decreased accordingly, but is still underdeveloped. The dietary problems that existed before 2015 mentioned in the study of Zhao et al. are still not fully resolved [5]. The consumption of fruits, vegetables, and milk and dairy products is still seriously less than the intake range recommended by the Chinese Nutrition Society. There is also an imbalance between urban and rural areas, and the dietary structure in rural areas needs to be improved.

Food consumption, in addition to dietary imbalances, shows clear regional imbalances. For example, in 2021, the per capita consumption of grains, edible oils and meat in Tibet was in the middle to upper levels of consumption, while the per capita consumption of other food categories was in the lowest level of consumption; thus, there was a seriously unbalanced food consumption structure.

Food security is defined as physical, social, and economic access for all people at all times to a sufficient quantity and quality of food that meets their dietary needs and food preferences for an active and healthy life in terms of variety, diversity, nutrition, and safety [21]. The issue of food consumption is an integral part of food security and the key to achieving the strategic goal of a healthy China is to enable the population to eat healthily. The above results show that the country’s dietary balance is not ideal. And food consumption is influenced by a number of external factors.

Many studies have examined the relationship between influencing factors such as income, urbanization, and food production and dietary health, and some studies have concluded that these factors have a positive impact on the improvement of dietary balance [16,22,23,24,25]. The effect of income is shown by the increase in consumption of animal-based foods, dairy products, fruits and vegetables with increasing income among residents, and in the difference in the degree of dietary balance between urban and rural residents. Urbanization also causes changes in residents’ lifestyles and dietary preferences. Urbanization promotes the upgrading of food consumption structure and nutritional intake by increasing residents’ awareness of nutrition and health, reducing labor intensity, decreasing market distance, and increasing the availability of nutritious food to urban and rural residents. A diverse food structure is an important foundation for the nutritional health of residents, and food production often influences local dietary preferences. For example, the southeastern coastal regions are rich in fishery products, and, accordingly, the consumption of fishery products in these regions is much higher than that in other regions. However, some scholars argue that the influence of these factors cannot be generalized. For example, You et al. argue that the effect of income on improved nutritional balance is weak, while other factors (e.g., food prices, aging), can easily offset the weak effect of income [26]. In response to the current food consumption situation, corresponding changes can be suggested. Firstly, there is a need to cultivate a balanced diet food consumption concept among residents, especially for those in rural areas of central and western China [27]. As urbanization and income levels increase in China, animal-based foods consumption will continue to increase, and the environmental pressure and health risks caused by an unreasonable dietary structure will continue to be exposed. Residents’ food consumption is closely related to dietary habits and traditional culture. Therefore, it is necessary to take into account the national situation and advocate that the public eat a variety of foods with diets consisting of mainly cereals, but also more vegetables, milk and dairy products, more animal-based foods, in moderation, and less salt and oil, according to the recommended standards of the Chinese Food Guide Pagoda (2022). Secondly, the food supply should be enriched to enhance the residents’ access to food of insufficient consumption. The prerequisite for ensuring that residents have access to sufficient food is to produce sufficient food. In terms of agricultural supply, we need to actively develop dairy production and promote the development of high-quality pastures. Through the development of new breeding methods, we can overcome the seasonal and geographical difficulties faced by agricultural production in order to increase the supply of under-consumed food such as vegetables and fruits. Agrotourism can also be developed to in turn influence the production and consumption of local foods [28]. There is an agglomeration effect in the consumption of food by our residents, and regions with high (low) consumption of a certain food will also have high (low) consumption in their surrounding areas. This feature can be used in the process of eliminating regional differences to improve the food consumption of neighboring regions through the optimization of the food consumption structure in one region.

Limitations exist in this paper. Constrained by the data in the yearbook, a more comprehensive analysis of all food groups and more influential factors in the dietary pagoda was not possible. The gradual diversification of food consumption will also bring about nutritional problems due to unreasonable dietary structure, that may produce a situation of overnutrition and high incidence of chronic diseases. In addition, the shift in consumption structure will bring certain challenges to food security and resources and the environment; the balance of food supply and demand and the allocation of environmental resources are topics that need attention in the future. Future studies can also be conducted in more depth for the regions with the most severe dietary imbalance.

## 5. Conclusions

Using publicly available food consumption data, this paper reviewed and analyzed food consumption in China at the macro level over the past seven years, discussed changes and problems in food consumption, and provided suggestions for optimization to inform future related research and policy formulation. From 2015 to 2021, the food consumption structure of China’s residents developed towards a balanced diet. However, it is still underdeveloped, with the consumption of fruits, vegetables, milk and dairy products not meeting the recommended intake range of the Chinese Dietary Guidelines being the most serious shortfall. Food consumption exhibits significant regional imbalances in the context of dietary imbalances and is influenced by urbanization, disposable income, and production. There is a clustering effect of food consumption in China, and more attention should be paid to regions where particular foods are under-consumed.

## Figures and Tables

**Figure 1 foods-12-01997-f001:**
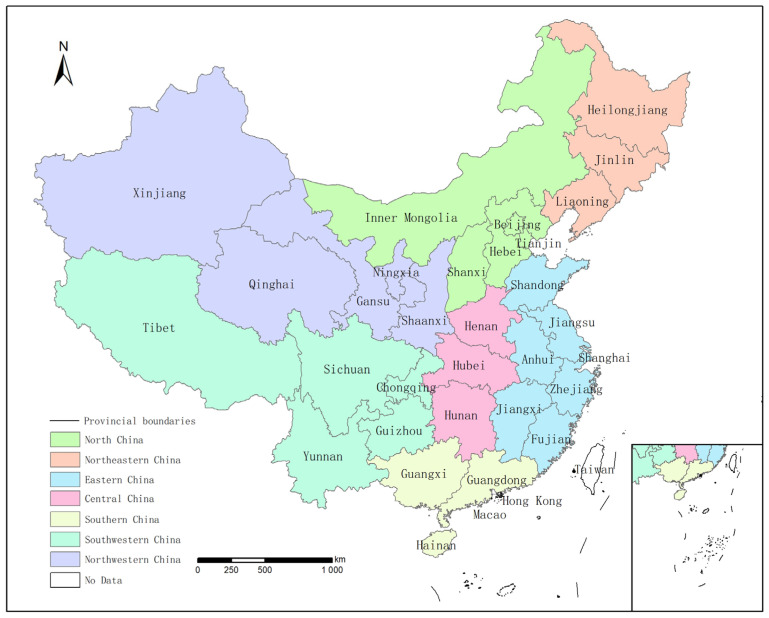
Seven natural geographic regions.

**Figure 2 foods-12-01997-f002:**
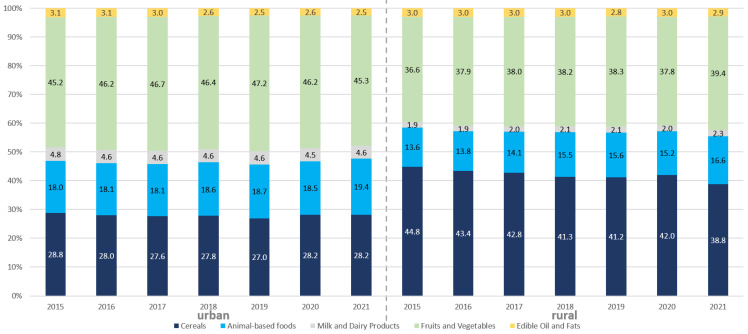
The structure of food consumption, 2015–2021.

**Figure 3 foods-12-01997-f003:**
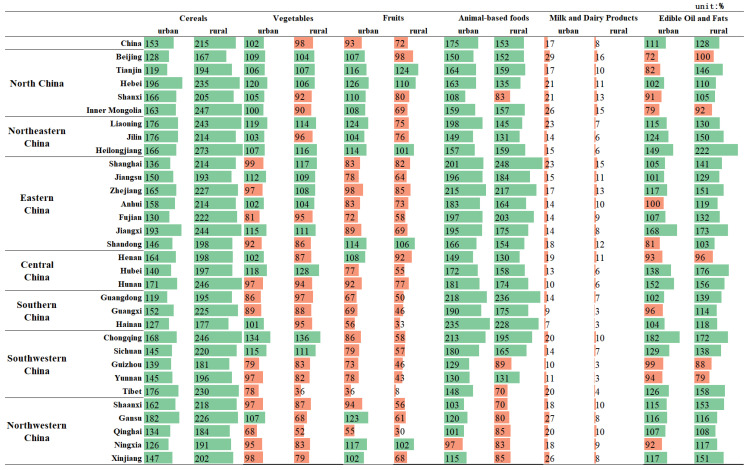
Percentage of food consumption relative to the minimum recommended by the Chinese Food Guide Pagoda, 2021. (The orange color bar indicates that the recommended criteria are not met, while the green color bar indicates that the recommended criteria are met).

**Figure 4 foods-12-01997-f004:**
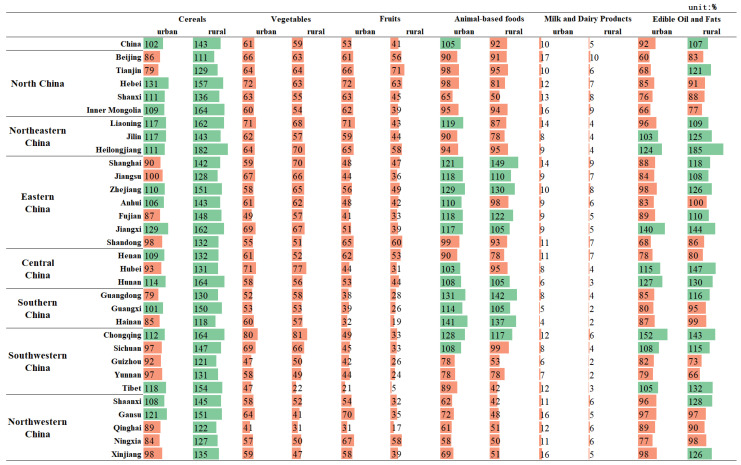
Percentage of food consumption relative to the maximum recommended by the Chinese Food Guide Pagoda, 2021. (The orange color bar indicates that the recommended criteria are not met, while the green color bar indicates that the recommended criteria are met).

**Figure 5 foods-12-01997-f005:**
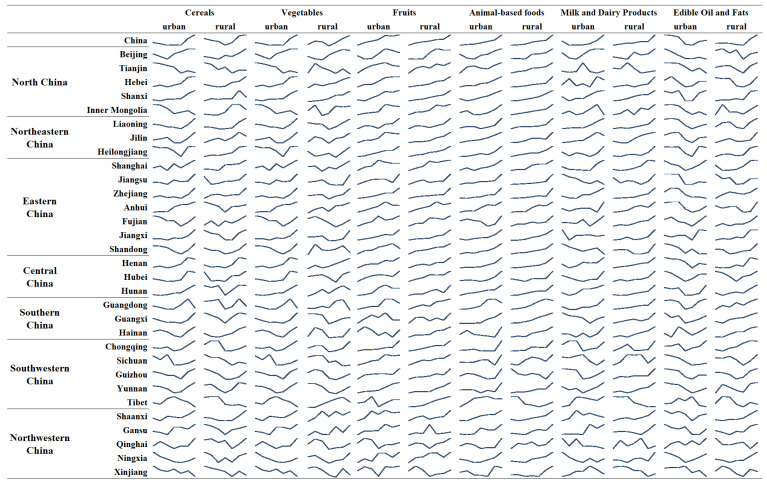
Trends in percentage of food consumption relative to the minimum recommended by Chinese Food Guide Pagoda, 2015–2021.

**Figure 6 foods-12-01997-f006:**
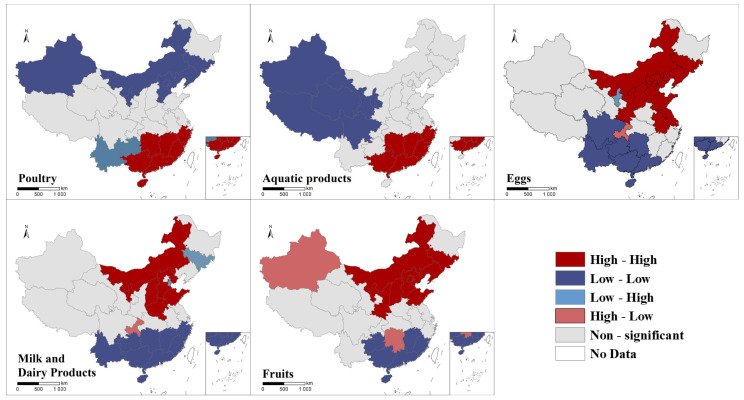
Spatial autocorrelation of food consumption of urban residents in China, 2021.

**Figure 7 foods-12-01997-f007:**
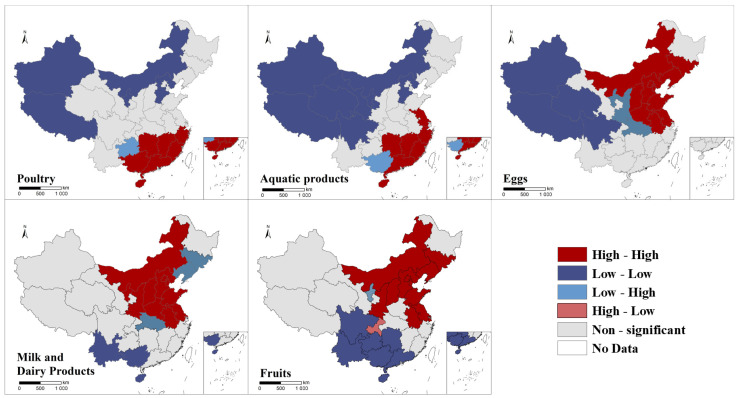
Spatial autocorrelation of food consumption of rural residents in China, 2021.

**Figure 8 foods-12-01997-f008:**
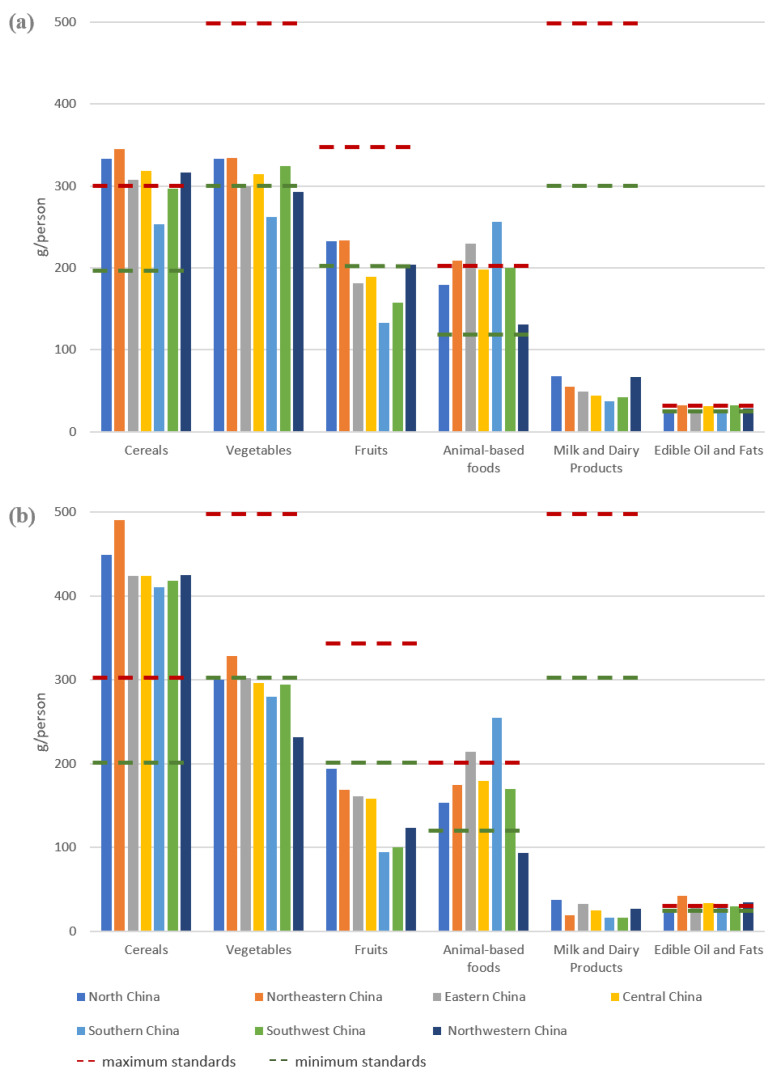
Average daily consumption of food in the seven geographic regions, 2021. (**a**) urban; (**b**) rural.

**Figure 9 foods-12-01997-f009:**
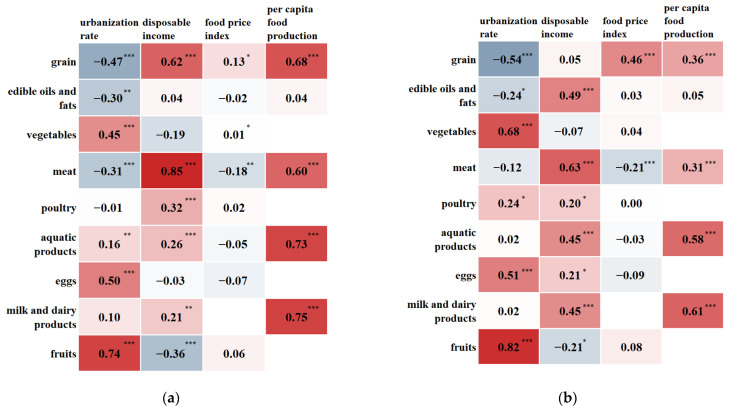
Regression analysis of the factors influencing food consumption (The numbers in the graph represent the standardized regression coefficients) (**a**) urban, * *p* < 0.05, ** *p* < 0.01, *** *p* < 0.001; (**b**) rural, * *p* < 0.05, *** *p* < 0.001.

**Table 1 foods-12-01997-t001:** Daily recommended values for various food groups in the Chinese Food Guide Pagoda (2022).

Category	Min	Max
cereals	200 g	300 g
—whole grains and mixed legumes	50 g	150 g
tubers	50 g	100 g
vegetables	300 g	500 g
fruits	200 g	350 g
animal-based foods	120 g	200 g
—Aquatic products at least twice a week —one egg per day	/	/
milk and dairy products	300 g	500 g
Soybeans and nuts	25 g	35 g
salt	/	5 g
edible oils and fats	25 g	30 g

**Table 2 foods-12-01997-t002:** Selection of variables.

Consumption Data	Influencing Factors
Ln (Y_1_)	Ln (X_1_), Ln (X_2_), Ln (X_3_), Ln (X_4_)
Ln (Y_2_)	Ln (X_1_), Ln (X_2_), Ln (X_3_), Ln (X_4_)
Ln (Y_3_)	Ln (X_1_), Ln (X_2_), Ln (X_3_)
Ln (Y_4_)	Ln (X_1_), Ln (X_2_), Ln (X_3_), Ln (X_4_)
Ln (Y_5_)	Ln (X_1_), Ln (X_2_), Ln (X_3_)
Ln (Y_6_)	Ln (X_1_), Ln (X_2_), Ln (X_3_), Ln (X_4_)
Ln (Y_7_)	Ln (X_1_), Ln (X_2_), Ln (X_3_)
Ln (Y_8_)	Ln (X_1_), Ln (X_2_), Ln (X_4_)
Ln (Y_9_)	Ln (X_1_), Ln (X_2_), Ln (X_3_)

**Table 3 foods-12-01997-t003:** Spatial autocorrelation test for per capita consumption, 2021.

	Urban		Rural	
	Moran’s *I*	*p* Value	Moran’s *I*	*p* Value
Poultry	0.400	0.030	0.392	0.003
Aquatic Products	0.376	0.001	0.372	0.001
Eggs	0.461	0.001	0.435	0.001
Milk and Dairy Products	0.330	0.001	0.278	0.002
Fruits	0.314	0.010	0.322	0.001

## Data Availability

The data are included in the article.

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
