# Peer review of "The Reasonableness and Spatial Differences of the Food Consumption Structure of Urban and Rural Residents in China, 2015–2021"

_foods, 2023, doi:10.3390/foods12101997_

Round 1

Reviewer 1 Report

The authors are thanked for the interesting research topic. The authors highlighted the estimation of food consumption in a large number of Chinese provinces and investigated the deviation of consumption from Based on food consumption data of residents from 31 provinces in China from 2015-2021, 10

the study analyzes the deviation of food consumption from target nutrition and spatial target nutrition. Although the manuscript is very well written, I would like to point out some suggestions for correction to improve the quality.

The title and abstract are completely fine. However, I am of the opinion that the introductory part should be expanded and even a literature review chapter should be created. Very little similar research has been presented, so that we can see the importance of this manuscript in a wider context. The methodology is well explained, and the statistical results are presented quite clearly through figures and tables, very legible with high resolution. I believe that nothing should be changed in that segment.

Also in the discussion part, it is necessary to mention some similar researches and results that the authors refer to and emphasize the goal of their research and comparative results. We would certainly see the importance of the obtained results. The discussion is very concise. It is necessary to expand, as well as concluding considerations. In the part of the conclusion, explain the theoretical and applied significance of the research, as well as highlight the future implications.

Suggested reference

 Development of the Concept of Sustainable Agro-Tourism

Destinations—Exploring the Motivations of Serbian Gastro-Tourists. Sustainability, 2023, 15, 2839. https://doi.org/10.3390/su15032839

Author Response

Response to Reviewer 1 Comments

Dear Reviewer:

We would like to thank you for your constructive and helpful reviews and useful comments and suggestions. Those comments are all valuable and very helpful for revising and improving our paper, as well as the important guiding significance to our researches. We have revised the manuscript accordingly and detailed corrections are listed below point by point. The revised manuscript has been proofread by all authors to improve the readability and avoid possible grammatical errors. We highlighted the revised words/sections in the revised manuscript in red color to track the changes/additions we've made. The main corrections in the paper and the responds to the comments are as following:

Point 1: The title and abstract are completely fine. However, I am of the opinion that the introductory part should be expanded and even a literature review chapter should be created. Very little similar research has been presented, so that we can see the importance of this manuscript in a wider context.

Response 1: Thanks for your suggestion. We have added related information in the p. 2, lines 53-67.

Revised text: “Many scholars have conducted certain studies on the food consumption, which mainly include the following aspects: firstly, studies on the changes in food consumption, which often use long time series of food consumption data to elaborate the trends and patterns of changes in the food consumption structure of China at the macro level [1, 5]. The second type of studies focuses on forecasting the future food demand and supply for the whole country or a key region by methods such as formula calculations [14, 15]. Other studies are mainly based on a nutritional perspective and investigate the characteristics of food consumption and dietary nutrition status and the differences be-tween them by methods such as household surveys or macro statistics [2]. The fourth category of literature examines the interactions between food consumption and various types of resources and environments. The last category studies the socioeconomic factors that have an impact on the residents’ food consumption, such as income, urbanization level, food price index, and food production [16]. These factors can in turn influence the dietary structure of the residents by changing their health consciousness, dietary preferences, ability to access food, and ease of access to food.”

Point 2: Also in the discussion part, it is necessary to mention some similar researches and results that the authors refer to and emphasize the goal of their research and comparative results. We would certainly see the importance of the obtained results. The discussion is very concise. It is necessary to expand, as well as concluding considerations.

Response 2: Thanks for your suggestion. We have added related information in the p 11-12.

Revised text: 1) line 278-282, “Hou et al. studied the changes in dietary consumption of the Chinese population between 1980 and 2021, which were mainly characterized by a decline in direct consumption of cereals, a rapid rise in consumption of animal products, and a continuous increase in vegetables and fruits [1]. The results presented in this paper are consistent with them.”

2) line 291-293, “The dietary problems that existed before 2015 mentioned in the study of Zhao et al. are still not fully resolved [5].”

3) line 309-326, “Many studies have examined the relationship between influencing factors such as income, urbanization, and food production and dietary health, and some studies have concluded that these factors have a positive impact on the improvement of dietary balance [16,22-25]. The effect of income is shown by the increase in consumption of animal-based foods, dairy products, fruits and vegetables with the increase of residents' income, and the difference in the degree of dietary balance between urban and rural residents. Urbanization also causes changes in residents' lifestyles and dietary preferences. Urbanization promotes the upgrading of food consumption structure and nutritional intake by increasing residents' awareness of nutrition and health, reducing labor intensity, decreasing market distance, and increasing the availability of nutritious food to urban farmers or rural residents. A diverse food structure is an important foundation for the nutritional health of residents, and food production often influences local dietary preferences. For example, the southeastern coastal regions are rich in fishery products, and accordingly, the consumption of fishery products in these regions is much higher than that in other regions. However, some scholars argue that the in-fluence of these factors cannot be generalized. For example, YOU et al. argue that the effect of income on improved nutritional balance is weak, while other factors (e.g., food prices, aging), can easily offset the weak effect of income [26].”

Point 3: In the part of the conclusion, explain the theoretical and applied significance of the research, as well as highlight the future implications.

Response 3: Thanks for your suggestion. We have added related information in the p 12.

Revised text: “Using publicly available food consumption data, this paper reviews and analyzes food consumption in China at the macro level over the past seven years, discusses changes and problems in food consumption, and makes suggestions for optimization to inform future related research and policy formulation.”

Point 4: Suggested reference: Development of the Concept of Sustainable Agro-Tourism Destinations—Exploring the Motivations of Serbian Gastro-Tourists. Sustainability, 2023, 15, 2839. https://doi.org/10.3390/su15032839

Response 4: Thanks for your suggestion. This literature has been cited in the article. (reference28)

Revised text: “Agrotourism can also be developed to in turn influence the production and consumption of local foods [28].”

Furthermore, we added 8 references due to modification requirements and marked in red color in revised paper, and the added references are as follows.

  1. National Bureau of Statistics. Statistical division of urban and rural areas (approved by State letter [2008] No. 60). http://www.stats.gov.cn/sj/tjbz/gjtjbz/202302/t20230213_1902742.html.
  2. Xin, L.J.; Wang, J.Y.; Wang, L.X. Prospect of per capita grain demand driven by dietary structure change in China[J]. Resources Science, 2015,37(7):1347-1356.
  3. Ding, R.; Shi, W.J.; Lu, C.H.; Shi, X.L.; Deng, X.Z; Cui, J.Y. Future unbalanced-trends of grain supply and demand on the Tibetan Plateau [J]. Journal of Cleaner Production, 2022(367):132993.
  4. Wang, L.E., Guo, J.X.; Ling, F.; Luo, Y.H.; Zhang, X.Z.; Fan, Y.Z.; Cheng, S.K. The structure and characteristics of resident food consumption in rural areas of the Tibetan Plateau: Taking Three-Rivers Region in Tibet as an example[J]. Acta Geographica Sinica, 2021,76(9):2104-2117.
  5. Hou M.H. Study on the Influence of Rural Residents' Income Level and Dietary Knowledge on Food Consumption [D]. Huazhong Agricultural University, 2022.
  6. Li X.Y.; Zhang X.J. The Impact of Income and Agricultural Production Categories on Nutrition Intake of Rural Residents in China[J]. Journal of Huazhong Agricultural University (Social Science Edition), 2020(04):37-49.
  7. You J.; Imai, K.S.; Gaiha, R. Declining Nutrient Intake in a Growing China: Does Household Heterogeneity Matter?[J]. World Development, 2016,77:171-191.
  8. Vukolić. D.; Gajić. T.; Petrović. M.D.; Bugarčić, J.; Spasojević, A. Veljović, S.; Vuksanović, N.; Bugarčić, M.; Zrnić, M.; Knežević, S.; Rakić, S.R.; Drašković, B.D.; Petrović, T. Development of the Concept of Sustainable Agro-Tourism Destinations-Exploring the Motivations of Serbian Gastro-Tourists[J]. Sustainability, 2023,15(3):2839.

We have made every effort to revise our manuscript in light of the comments and have marked the revisions in red.

We sincerely thank the editors and reviewers for their enthusiastic work and hope that these revisions will be acknowledged.

Once again, thank you very much for your comments and suggestions.

Best regards.

The authors

Reviewer 2 Report

I thank the editors for the opportunity to review this study, moreover I would also like to congratulate the authors for the made effort in their study. The present manuscript by Lian et al. is entitled: “The Reasonableness and Spatial Differences of the Food Consumption Structure of Urban and Rural Residents in China, 2015-2021”. The authors want to explore the main characteristics and problems of the current food consumption and nutritional development of Chinese urban and rural residents and the influencing factors based on the theory of dietary balance, analyzes the differences between geographical areas, and sees whether the problems in food consumption have been solved or improved in the last seven years compared to those in the past. Please consider the following commentaries to improve the manuscript.

Introduction

- Authors should clarify the concept of Chinese urban and rural areas.

- Authors should discuss, considering available literature, how the following factors: “urbanization rate”, “disposable income”, “food price index”, “per capita food production” influence food consumption. This is particularly important as these factors are reported in section 3.5. Influencing factors of food consumption (Results section).

- Authors should better describe the Chinese Food Guide Pagoda (2022), its principles and its food groups. This is particularly important as the food intake recommended in Chinese Food Guide Pagoda (2022) was used to analyze the deviation and changes in food consumption of the population from the recommended standards.

Materials and methods

- Authors should describe the aim of the “China Yearbook of Household Survey”, framed as Household Income and Expenditure Surveys that provide information on people's living conditions and income/expenditure patterns.

Results

- p. 5, lines 149-151: Can authors please explain the main difference between “eat well” and “eat healthily”?

- For Figures 4 and 5 we ask authors to add another column (Region) with the corresponding region for each Chinese province.

- Figures 4 and 5 are Tables; we ask authors to update accordingly in the manuscript.

- p. 7, line 178: Typo error, there is a full stop missing between “the Figure 5 The graph”.

- Figure 5 (p. 7): We ask authors to add another column (Region) with the corresponding region for each Chinese province.

- Figures 6 and 7: i) they have the same title (please confirm and correct); ii) the scales of reference included in these figures are not clear.

- For the section “3.4. Regional food consumption” we ask authors to expose the main results of the average daily consumption of “milk and dairy products”.

Discussion

The authors should put more effort into the discussion section and focus on discussing their results with the available literature.

Author Response

Response to Reviewer 2 Comments

Dear Reviewer:

Firstly, we would like to thank you for your constructive and helpful reviews and useful comments and suggestions. Those comments are all valuable and very helpful for revising and improving our paper, as well as the important guiding significance to our researches. We have revised the manuscript accordingly and detailed corrections are listed below point by point. The revised manuscript has been proofread by all authors to improve the readability and avoid possible grammatical errors. We highlighted the revised words/sections in the revised manuscript in red color to track the changes/additions we've made. The main corrections in the paper and the responds to the comments are as following (Please see attachment for pictures):

Point 1: Introduction - Authors should clarify the concept of Chinese urban and rural areas.

Response 1: Thanks for your suggestion. We have added related information in the p. 1, lines 36-40.

Revised text: “And it is often statistically divided into two parts: urban and rural, with urban refer-ring to the residential councils and other areas connected to the actual construction of municipal, district, and county government sites in municipal districts and unincorporated cities, and rural being the areas outside of towns [8].”

Point 2: Introduction - - Authors should discuss, considering available literature, how the following factors: “urbanization rate”, “disposable income”, “food price index”, “per capita food production” influence food consumption. This is particularly important as these factors are reported in section 3.5. Influencing factors of food consumption (Results section).

Response 2: Thanks for your suggestion. We have added related information in the p. 2, lines 63-67. For a more specific discussion, please see the discussion section.

Revised text: “The last category studies the socioeconomic factors that have an impact on the residents’ food consumption, such as income, urbanization level, food price index, and food production [16]. These factors can in turn influence the dietary structure of the residents by changing their health consciousness, dietary preferences, ability to access food, and ease of access to food.”

Point 3: Introduction - Authors should better describe the Chinese Food Guide Pagoda (2022), its principles and its food groups. This is particularly important as the food intake recommended in Chinese Food Guide Pagoda (2022) was used to analyze the deviation and changes in food consumption of the population from the recommended standards.

Response 3: Thanks for your suggestion. We have added related information in the p. 2, lines 48-52.

Revised text: “In terms of human health, the Chinese Dietary Guidelines is a visual representation of dietary health. It follows the principle of balanced diet and divides the pagoda into 5 layers with different sizes to reflect the 5 major food groups and the amount of food. These 5 major food groups are cereals and potatoes, vegetables and fruits, livestock, poultry, fish, eggs and milk, soybeans and nuts, and cooking oil and salt.”

Point 4: Materials and methods - Authors should describe the aim of the “China Yearbook of Household Survey”, framed as Household Income and Expenditure Surveys that provide information on people's living conditions and income/expenditure patterns.

Response 4: Thanks for your suggestion. We have added related information in the p. 2, lines 80-83.

Revised text: “The China Yearbook of Household Survey is a national survey of household income and expenditure and living conditions in order to provide a comprehensive, accurate and timely understanding of the income, consumption and other living conditions and quality of life of urban and rural residents nationwide and in all regions.”

Point 5: Results - p. 5, lines 149-151: Can authors please explain the main difference between “eat well” and “eat healthily”?

Response 5: Thanks for your suggestion. “Eat well” means that the residents can meet their needs in terms of quantity of food, while “eat healthily” means that the nutritional needs are met. This expression is really inaccurate, and has been revised in p. 5, line 175.

Revised text: “However, from the perspective of nutrition and health, residents should not only eat enough, but also eat healthily.”

Point 6: Results - For Figures 4 and 5 we ask authors to add another column (Region) with the corresponding region for each Chinese province.

Response 6: Thanks for your suggestion. We have added another column in the table 3 and table 4.

Revised table 3:

Table 3. Percentage of food consumption relative to the minimum recommended by the Chinese Food Guide Pagoda, 2021

Revised table 4:

Table 4. Percentage of food consumption relative to the maximum recommended by the Chinese Food Guide Pagoda, 2021

Point 7: Results - Figures 4 and 5 are Tables; we ask authors to update accordingly in the manuscript.

Response 8: Thanks for your suggestion. We have revised accordingly, please see point 6.

Point 8: Results - p. 7, line 178: Typo error, there is a full stop missing between “the Figure 5 The graph”.

Response 8: Thanks for your suggestion. We have revised in the p. 7, line 202.

Revised text: “The trend of food consumption as a percentage of the dietary pagoda from 2015-2021 is shown in Figure Table 5. It shows…”

Point 9: Results - Figure 5 (p. 7): We ask authors to add another column (Region) with the corresponding region for each Chinese province.

Response 9: Thanks for your suggestion. We have added another column in the table 5.

Revised table 5:

Table 5. Trends in percentage of food consumption relative to the minimum recommended by Chinese Food Guide Pagoda, 2015-2021

Point 10: Results - Figures 6 and 7: i) they have the same title (please confirm and correct); ii) the scales of reference included in these figures are not clear.

Response 10: Thanks for your suggestion. We have revised accordingly in the figure 3 and figure 4.

Revised figure 3:

Figure 3. Spatial autocorrelation of food consumption of urban residents in China, 2021

Revised figure 4:

Figure 4. Spatial autocorrelation of food consumption of rural residents in China, 2021

Point 11: Results - For the section “3.4. Regional food consumption” we ask authors to expose the main results of the average daily consumption of “milk and dairy products”.

Response 11: Thanks for your suggestion. We have added related information in the p. 10, lines 256-258.

Revised text: “Milk and dairy products are well below the lower limit of the recommended range in both urban and rural areas or in either region.”

Point 12: Discussion - The authors should put more effort into the discussion section and focus on discussing their results with the available literature.

Response 12: Thanks for your suggestion. We have added related information in the p 11.

Revised text: 1) line 278-282, “Hou et al. studied the changes in dietary consumption of the Chinese population between 1980 and 2021, which were mainly characterized by a decline in direct consumption of cereals, a rapid rise in consumption of animal products, and a continuous increase in vegetables and fruits [1]. The results presented in this paper are consistent with them.”

2) line 291-293, “The dietary problems that existed before 2015 mentioned in the study of Zhao et al. are still not fully resolved [5].”

3) line 309-326, “Many studies have examined the relationship between influencing factors such as income, urbanization, and food production and dietary health, and some studies have concluded that these factors have a positive impact on the improvement of dietary balance [16,22-25]. The effect of income is shown by the increase in consumption of animal-based foods, dairy products, fruits and vegetables with the increase of residents' income, and the difference in the degree of dietary balance between urban and rural residents. Urbanization also causes changes in residents' lifestyles and dietary preferences. Urbanization promotes the upgrading of food consumption structure and nutritional intake by increasing residents' awareness of nutrition and health, reducing labor intensity, decreasing market distance, and increasing the availability of nutritious food to urban farmers or rural residents. A diverse food structure is an important foundation for the nutritional health of residents, and food production often influences local dietary preferences. For example, the southeastern coastal regions are rich in fishery products, and accordingly, the consumption of fishery products in these regions is much higher than that in other regions. However, some scholars argue that the in-fluence of these factors cannot be generalized. For example, YOU et al. argue that the effect of income on improved nutritional balance is weak, while other factors (e.g., food prices, aging), can easily offset the weak effect of income [26].”

Furthermore, we added 8 references due to modification requirements and marked in red color in revised paper, and the added references are as follows.

  1. National Bureau of Statistics. Statistical division of urban and rural areas (approved by State letter [2008] No. 60). http://www.stats.gov.cn/sj/tjbz/gjtjbz/202302/t20230213_1902742.html.
  2. Xin, L.J.; Wang, J.Y.; Wang, L.X. Prospect of per capita grain demand driven by dietary structure change in China[J]. Resources Science, 2015,37(7):1347-1356.
  3. Ding, R.; Shi, W.J.; Lu, C.H.; Shi, X.L.; Deng, X.Z; Cui, J.Y. Future unbalanced-trends of grain supply and demand on the Tibetan Plateau [J]. Journal of Cleaner Production, 2022(367):132993.
  4. Wang, L.E., Guo, J.X.; Ling, F.; Luo, Y.H.; Zhang, X.Z.; Fan, Y.Z.; Cheng, S.K. The structure and characteristics of resident food consumption in rural areas of the Tibetan Plateau: Taking Three-Rivers Region in Tibet as an example[J]. Acta Geographica Sinica, 2021,76(9):2104-2117.
  5. Hou M.H. Study on the Influence of Rural Residents' Income Level and Dietary Knowledge on Food Consumption [D]. Huazhong Agricultural University, 2022.
  6. Li X.Y.; Zhang X.J. The Impact of Income and Agricultural Production Categories on Nutrition Intake of Rural Residents in China[J]. Journal of Huazhong Agricultural University (Social Science Edition), 2020(04):37-49.
  7. You J.; Imai, K.S.; Gaiha, R. Declining Nutrient Intake in a Growing China: Does Household Heterogeneity Matter?[J]. World Development, 2016,77:171-191.
  8. Vukolić. D.; Gajić. T.; Petrović. M.D.; Bugarčić, J.; Spasojević, A. Veljović, S.; Vuksanović, N.; Bugarčić, M.; Zrnić, M.; Knežević, S.; Rakić, S.R.; Drašković, B.D.; Petrović, T. Development of the Concept of Sustainable Agro-Tourism Destinations-Exploring the Motivations of Serbian Gastro-Tourists[J]. Sustainability, 2023,15(3):2839.

We have made every effort to revise our manuscript in light of the comments and have marked the revisions in red.

We sincerely thank the editors and reviewers for their enthusiastic work and hope that these revisions will be acknowledged.

Once again, thank you very much for your comments and suggestions.

Best regards.

The authors
